# Laser Polishing Die Steel Assisted by Steady Magnetic Field

**DOI:** 10.3390/mi13091493

**Published:** 2022-09-08

**Authors:** Zhenyu Zhao, Junyong Zeng, Zhouyi Lai, Jie Yin, Ting Guo

**Affiliations:** 1School of Sino-German Robotics, Shenzhen Institute of Information Technology, Shenzhen 518029, China; 2College of Mechanical and Control Engineering, Shenzhen University, Shenzhen 518060, China; 3School of Mechanical Engineering, Xiangtan University, Xiangtan 411105, China

**Keywords:** die steel, steady magnetic field, laser polishing, numerical model, molten pool

## Abstract

To improve the surface roughness of SKD61 die steel and reduce the secondary overflow of the molten pool, a steady magnetic field-assisted laser polishing method is proposed to study the effect of steady magnetic field on the surface morphology and melt pool flow behavior of SKD61 die steel. Firstly, a low-energy pulsed laser is used for the removal of impurities from the material surface; then, the CW laser, assisted by steady magnetic field, is used to polish the rough surface of SKD61 die steel to reduce the material surface roughness. The results show that the steady magnetic field-assisted laser polishing can reduce the surface roughness of SKD61 die steel from 6.1 μm to 0.607 μm, which is a 90.05% reduction compared with the initial surface roughness. Furthermore, a multi-physical-field numerical transient model involving heat transfer, laminar flow and electromagnetic field is established to simulate the flow state of the molten pool on the surface of the SKD61 die steel. This revealed that the steady magnetic field is able to inhibit the secondary overflow of the molten pool to improve the surface roughness of SKD61 slightly by reducing the velocity of the molten pool. Compared with the molten pool depth obtained experimentally, the molten pool depth simulation was 65 μm, representing an error 15.0%, thus effectively demonstrating the accuracy of the simulation model.

## 1. Introduction

As a medium-carbon, high-chromium alloy tool steel, SKD61 steel has a lot of advantages, such as high hardenability, high toughness, excellent cracking resistance and good dimensional stability [1,2]. It is a kind of steel formed by adding alloying elements on the basis of carbon steel, implementing the standard GB/T1299-2014. SKD61 steel is the most widely used and representative hot work die steel, and is widely used in the field of hot stamping dies and hot extrusion dies [3,4]. The surface roughness affects the accuracy of the die. At present, the improvement of the mold surface accuracy is mainly based on the traditional contact manual mechanical polishing. With laser polishing, as a non-contact metal surface smoothing technology, the surface roughness can reach micron or even nanometer scale.

Laser polishing has unique advantages such as no contact, no raw material loss, no environmental pollution, and selective polishing compared with the traditional polishing methods [5]. In addition, it can not only polish almost all metals [6,7,8,9,10,11,12,13,14], but can also be used to polish nonmetallic materials such as glass [13], ceramics [14,15,16], and optical materials such as KDP crystals [17]. In addition, laser polishing can also effectively reduce the surface roughness of 3D-printed products through the rapid development of laser polishing technology. Ukar E. et al. [18] studied the effect of energy density on surface roughness in two different lasers: CO_2_ laser and semiconductor laser. The experimental results showed that the roughness of the sample surface was reduced to 0.5 μm when polished by CO_2_ laser polishing, representing a reduction of up to 90%. Bhaduri et al. [19] studied the effect of energy density on the surface characteristics of 3D-printed 316 L stainless steel. It was found that when the energy density was between 18 J/cm^2^ and 20 J/cm^2^, the surface was seriously ablated, resulting in surface over-melting; when the energy density was below 13 J/cm^2^, it produced a shiny metal surface with a better surface polishing effect than when the energy density was too high. Ramos et al. [20] believed that there were two different states of surface modification under laser polishing: surface shallow melting (SSM (Figure 1a)) and surface over-melting (SOM (Figure 1b)). The short residence time of the laser beam and the position of the melting area above the peak waist results in the shallow melting of the surface. Then, due to the gravity and surface tension of the material and other forms of force, the melting part of the material fills the valley from top to bottom, resulting in a smooth surface. As for surface over-melting, the molten material solidifies before it stops vibrating due to the excessive laser energy applied to the sample surface, which leads to the formation of a new uneven surface. Zhang et al. [21] coupled heat transfer and laminar flow to establish a two-dimensional mathematical model to reveal the flow behavior of molten pools. Zhang et al. [12] studied the free surface evolution of laser polishing under the condition of a moving laser beam by means of numerical simulation and an experiment. Comparing the experimental results with the simulation results, it was found that the differences for molten pool depth and surface roughness were 7.5% and 7.4%, with an error rate of less than 8%. Thus, the test results were in good agreement with the simulation results. Wang et al. [22] proposed a multi-physical-field coupling simulation model with the help of steady magnetic field-assisted laser melting, taking into account the heat transfer, convection, phase transition, electromagnetic force and surface morphology of the molten pool. To study the effect of magnetic field-assisted laser polishing, Xiao et al. [23] used magnetic field-assisted laser polishing of S136H. The experimental results indicated that the surface roughness of the sample could be reduced by 91%, while the surface hardness of the sample could be increased by 112% at a magnetic field strength of 0.4 T, thus greatly improving the sample surface properties.

Although scholars have performed extensive research on laser polishing technology, numerical simulation and material properties, there are few systematic studies on compound field laser polishing. At present, there is a lack of in-depth research on its experimental and numerical simulation.

Therefore, this paper studies in detail the complex hydrodynamic behavior and the evolution mechanism of the surface morphology in the process of steady magnetic field-assisted laser polishing. In addition, a two-dimensional axisymmetric model is established by coupling heat transfer, laminar flow, and magnetic field in order to study the temperature distribution and velocity of the molten pool and the evolution of the free surface. Finally, the accuracy of the model is verified on the basis of experiments, and the inhibitory effect of steady magnetic field on the molten pool is discussed.

## 2. Materials and Methods

### 2.1. Experimental Materials

In this paper, the magnetic field-assisted CW laser polishing test was carried out on the EDM surface of SKD61 die steel with initial Ra = 6.1 μm. The basic chemical composition is shown in Table 1. The material was not heat treated prior to testing. The oil and oxide film of the sample surface was removed by pulsed laser, then cleaned with alcohol to reduce the influence of controllable environmental factors on the test.

### 2.2. Experimental Equipment

The experimental equipment is a multimode CW fiber laser (MFSC-1000W) with the wavelength of 1064 nm produced by Chuangxin Laser Co., Ltd., Shenzhen, China and the power of the laser is in the range of 150 W–1000 W. In addition, the laser mirrors are SDL-F20PR0-3 made by Fretak Laser Technology Co., Ltd., Jiangsu, China.

The maximum scanning range is 600 × 600 mm^2^, the focal length is 720 mm, the zoom range of the laser is 60 mm, and the diameter of the laser spot in the focal position is 0.3 mm. In this paper, all polishing tests are performed at a distance of 720 mm between the sample surface and the bottom of the oscillator without considering the amount of scattered focus. The energy distribution of continuous wave laser beam is flat-topped beam.

The principle of the laser polishing test equipment and the schematic diagram are shown in Figure 1. The device consists of a laser generator, a scanning mirror (dynamic focusing system), a beam expander (model: 2.8-355-200 m, JGZ Optical Technology Co., Ltd., Shenzhen, China), a closed cavity, an inert gas bottle, a water cooler, and a two-dimensional numerical control platform. The function of the scanning mirror is to adjust the laser beam position to ensure that the laser illuminates on the surface of the sample. To prevent the oxidation of the sample surface during laser polishing, argon is used as a protective gas, which will not react with the material. The two-dimensional CNC platform is used to realize the deflection or rotation of the sample.

Sample sizes of 70 × 20 × 5 mm^3^, 70 × 10 × 5 mm^3^ and 10 × 10 × 5 mm^3^ are used in magnetic field-assisted laser polishing, and the region of laser polishing is 9 × 9 mm^2^. The permanent magnet is made of NdFeBN52 strong magnet with a size of 80 × 10 × 10 mm^3^, which is used to provide a stable magnetic field for the molten pool. The container of the permanent magnet is made of aluminum and the size is 110 × 33 × 18 mm^3^. To produce a higher flux density, a group of magnets consisting of two magnets is arranged on both sides of the sample, as shown in Figure 2. Fill the container with water to cool the magnet and prevent demagnetization of the permanent magnet at high temperature. The magnetic field intensity can be determined by controlling the sample size and the distance between the poles and the number of magnets. The magnetic field intensity is measured using a Tesla meter (model: PFX-035, Lida Magnetic Technology Co., Shenzhen, China).

### 2.3. Experimental Method

To explore the effects of energy density and magnetic field intensity (MFI) on the surface morphology, the L_16_(4^4^) orthogonal experiment was designed to reduce the surface roughness of sample.

The experimental parameters of magnetic field-assisted continuous-wave laser polishing of SKD61 die steel are shown in Table 2, where MFI is the magnetic field intensity (T). The range of laser power (P) is 150 W–195 W with 15 W increments, the range of scanning speed (V) is 30–60 mm/s with 10 mm/s increments, the range of scanning spacing (D) is 0.3–0.9 mm with 0.2 mm increments, and the increment of magnetic field intensity (MFI) is 0.3 T in the range of 0–0.9 T. The polishing tests are carried out to study the effect of the laser power, scanning speed, scanning spacing and magnetic field intensity on the surface of SKD61 die steel.

### 2.4. Results and Discussion

Based on an analysis of the results of the magnetic field-assisted CW laser polishing experiments, it can be found that laser power, scanning speed, scanning spacing, and magnetic field intensity have a great influence on surface roughness.

As can be seen from Figure 3, with increasing energy density, the average surface roughness (Ra) decreases at first and then increases, showing a U-shaped curve. Figure 4 shows the mechanism of laser polishing. An energy density that is too high or too low is not conducive to reducing surface roughness, which can explain why the relationship between energy density and surface roughness in Figure 3 is U-shaped. When the energy density is higher than 1300 J/cm^2^, the residence time of heat on the sample surface increases, and too much heat accumulation leads to the melting of the material surface too deep. As a result, the molten material solidifies before the vibration stops, leading to irregular profile generation and the formation of the SOM phenomenon, as shown in Figure 4c. However, when the energy density is below 1200 J/cm^2^, this results in too little heat accumulation, and the material cannot be completely melted; therefore, the initial surface cannot be completely flattened, as shown in Figure 4a. When the energy density is 1200 J/cm^2^–1300 J/cm^2^, the melting region is above the peak waist, and then, due to forces in the form of gravity and the surface tension of the material, the melting part of the material fills the peak and valley from top to bottom, resulting in lower surface roughness and the formation of the SSM phenomenon, as shown in Figure 4b. When ED is increased to 1200 J/cm^2^, the best surface roughness is obtained. Throughout the experiment, the reduction rate of surface roughness ranges from 77.4% to 90%, indicating that the initial concave–convex surface has been effectively smoothed.

The three-dimensional surface morphology before and after magnetic field-assisted laser polishing was observed using a white light interferometer (model: BRUKER WYKO Contour GT-K, American)), as shown in Figure 5. Under the best polishing process, with a laser power of 180 W, a scanning speed of 50 mm/s, a scanning interval of 0.03 mm, and a magnetic field intensity of 0.3 T, the polished surface roughness of sample can be reduced to 0.607 μm, with a surface roughness reduction rate as high as 90.05%. In addition, the rough surface is further effectively smoothed due to the profile height difference being greatly improved. The two-dimensional surface curve of SKD61 die steel before and after laser polishing was obtained by Fourier filtering, as shown in Figure 6a, where h represents the height of the surface profile and X is the length of the surface profile. The black solid line represents the initial sample outline with Ram = 6.1 μm, and the red line represents the outline of sample 11 when polished with magnetic field. Under the action of the laser heat source, the temperature of the uneven surface above the melting point of the material, and the molten material is redistributed, resulting in a smooth surface being obtained from the molten pool flows. Then, the rapid solidification of the material surface achieves the effect of melting peak and valley filling, reducing the surface roughness and resulting in a smoother surface morphology being obtained. In the above characterization, the surface profile is analyzed only in the direction of the profile height, while the frequency analysis in the transverse space is ignored. The effect of laser polishing on the micro surface cannot be accurately observed. Therefore, power spectral density (PSD) is introduced to analyze the surface topography, where the horizontal axis represents the spatial frequency and the vertical axis represents the amplitude, as shown in Figure 6b. It can be seen that the two power spectral density arcs basically coincide when the spatial frequency is greater than 12 mm^−1^, and the PSD is attenuated to different degrees when the frequency is less than 12 mm^−1^, and the amplitude of the polished sample decreases significantly. Compared with the initial surface, the roughness of the polished surface is greatly reduced, and the roughness reduction rate reaches up to 90.05%.

## 3. Model Building

### 3.1. Model Hypothesis

To fully describe the evolution of the free surface and the flow behavior of the molten pool, a multi-physical-field numerical model is established by using four modules of the COMSOL multi-physical-field simulation software: fluid heat transfer (Heat Transfer in Fluids), laminar flow (Laminar Flow), electromagnetic field (Electromagnetic Field) and dynamic grid (Moving Mesh). The mass conservation equation, energy equation, Navier–Stokes equation and Maxwell equation are solved.

To simplify the practical problems and reduce the calculation time, this numerical simulation puts forward the following assumptions:The flow of molten metal is regarded as incompressible Newtonian laminar flow.The material has the characteristics of uniform distribution and is isotropic, and the thermal physical parameters of the material vary with temperature.The maximum temperature of laser polishing is lower than the boiling point of the material without considering the evaporation of the model.In the course of the experiment, because the sealing device is filled with argon protective gas, the chemical reaction between metal and air at high temperature is ignored.The gas domain is not considered in the modeling process. Due to the great differences in viscosity and density between the molten metal and the protective gas, the air domain can be ignored in order to reduce the calculation time.The flux density is uniformly distributed.

### 3.2. Control Equation

The numerical simulation of laser irradiation region is based on the general forms of continuity equation (Equation (1)), the momentum conservation equation (Navier–Stokes) (Equation (2)), the energy conservation equation (Equation (8)), and the Maxwell equations (Equations (10) and (11)) provided by the COMSOL multi-physics finite element software package.

The mass conservation equation is also called the continuity equation, and the formula is as follows:(1)$$\frac{\partial \rho }{\partial t}+\nabla \cdot \left(\rho u\right)=0$$

The ρ represents the mass density of SKD61 die steel, t represents time, u represents fluid velocity, and ∇ is the Hamiltonian operator.

The Lorentz force is added to the momentum conservation equation in the form of volume force, and the formula is as follows:(2)$$\rho \frac{\partial u}{\partial t}+\rho \left(u\cdot \nabla \right)u=\nabla \cdot \left[-pI+\eta \left(\nabla u+{\left(\nabla u\right)}^{T}\right)\right]+F_{v}$$
where p is the pressure on the molten pool, I the laser heat source with flat top distribution, η is the dynamic viscosity, T is the temperature, $F_{v}$ is the volume force, computed as the sum of buoyancy, Darcy damping, and Lorentz force, as shown below:(3)$$F_{v}=F_{Buoyancy}+F_{Darcy}+F_{Lorenz}$$

The three terms on the right are the buoyancy term, the Darcy damping term, and the Lorentz force term, respectively.
(4)$$F_{Buoyancy}=\rho \left(1-\beta \left(T-T_{ref}\right)\right)g$$
where g is the gravity constant, β is the coefficient of thermal expansion, and $T_{ref}$ is the reference temperature.
(5)$$F_{Darcy}=-c_{1}\frac{{\left(1-f_{L}\right)}^{2}}{f_{L}{}^{3}+c_{2}}u$$
where $F_{Darcy}$ is the source term of the equation and the friction dissipation of the paste region of the Karman–Kozney approximation derived from Darcy’s law. The main function is to slow down the flow velocity of the melt, in which $c_{1}$ is the large constant used to explain the morphology of the paste zone, which is determined by the morphology and size of dendrites in the microstructure of the material, and $c_{2}$ is a small constant to avoid being divided by 0. Here, values of $c_{1}$ = 10^12^, $c_{2}$= 0.001 are taken. $f_{L}$ is the volume fraction of the liquid phase, which is related to temperature and is used to describe the region between the melting zone and the matrix region, expressed as
(6)$$f_{L}=\left\{\begin{array}{l} 0\;\;\;\;\;\;\;\;\;\;\;\;\;\;\;\;T<T_{S},\\ \frac{T-T_{S}}{T_{L}-T_{S}}\;\;\;\;\;\;\;{\;T}_{S}\leq T\leq T_{L},\;\\ 1\;\;\;\;\;\;\;\;\;\;\;\;\;\;\;\;\;T>T_{L}.\; \end{array}\right.\;$$
where $T_{S}$ and $T_{L}$ represent the solidus and liquidus temperatures of SKD61 die steel, respectively. When the temperature is between the solid and liquid phases, the melt pool is a paste-like region with solid–liquid coexistence. When the temperature is higher than the liquidus temperature, the molten pool begins to form.
(7)$$F_{Lorenz}=j\times B=\sigma \left(E+u\times B\right)\times B$$
where $F_{Lorenz}$ is a Lorentz force term that couples the electromagnetic force in the molten pool with hydrodynamics, which is caused by the steady magnetic field provided by the permanent magnet and the movement of the molten conductive fluid. j is the current density, B is the magnetic flux density, E is the external electric field, the value of which is zero, and σ is the conductivity. The interaction between the external magnetic field and the moving molten metal droplets forms the Lorentz force, which acts in the opposite direction to the melt velocity, thus reducing the melt flow velocity. It couples with hydrodynamic and electromagnetic fields in the form of bulk forces in the fluid.

The energy conservation formula is as follows:(8)$$\rho C_{p}\frac{\partial T}{\partial t}+\rho C_{p}u\cdot \nabla T=\nabla \cdot \left(k\nabla T\right)+Q_{source}$$
where $C_{p}$ and k refer to constant-pressure heat capacity and thermal conductivity respectively. $Q_{source}$ is the heat source from the laser beam. For simplicity, the equivalent heat capacity is used to replace $C_{p}$. The latent heat of melting can be represented by the effect of temperature on it by modifying the specific heat capacity $C_{p}^{*}$, as follows:(9)$$C_{p}^{*}=L_{m}\frac{df_{L}}{dT}+C_{p}$$
where $L_{m}$ is the latent heat of melting.

The fixed Maxwell equations of magnetic field B and electric field E are as follows:(10)$$\nabla \times B=\mu_{0}j$$
(11)∇×E=0

The motion of conductive particles in the magnetic field is driven by the polishing speed, especially by the highly dynamic process in the molten pool, and thus the current density is induced according to Ohm’s law.

### 3.3. Computational Domain and Initial Model Establishment of the Model

Li Kai et al. [24] studied the effects of three different initial surface contours on laser polishing and found that the initial surface profile was an important factor affecting the final surface roughness of laser polishing. Therefore, the definition of the initial surface is very important in the simulation. In this model, the initial three-dimensional surface topography of the sample is measured using a white light interferometer (Figure 5a), and the two-dimensional profile is intercepted (Figure 7). The two-dimensional profile is filtered using a Fourier filter, and then the filtered curve is imported into COMSOL in the form of an interpolation function. The transient numerical model of free surface melting under the influence of surface tension and the Marangoni effect is established by using the COMSOL multi-physical-field simulation software package.

The size of the model area can be determined on the basis of the size of the molten pool obtained by the experiment. In addition, the length of the model is closely related to the width of the molten pool. However, the length of the molten pool is not easy to measure in the laser polishing experiment. Therefore, the length of the model domain is set with reference to the laser spot diameter (d). The beam diameter of the sample surface is 300 μm in the polishing experiments. To ensure that the top surface of the model can present a long enough polished surface profile, the length of the model is set to 3.33 times the diameter of the laser spot, which is 1000 μm. Figure 8 shows the geometry of the model. The width of the molten pool is 1000 μm, and the depth of the molten pool is 300 μm. This model domain not only meets the research needs of this paper, but also saves time when solving this multi-physical coupling model.

### 3.4. Material Properties

The material used in this study is SKD61 die steel, and the thermophysical parameters of the material are a function of temperature. Therefore, in order to make the simulation results closer to the real working conditions, the built-in interpolation function of COMSOL is used to represent the variation of thermophysical parameters with temperature. Figure 9 shows the curves of specific heat capacity, density, thermal conductivity, and dynamic viscosity of SKD61 die steel with temperature.

The prepared SKD61 tool steel (4Cr5MoSiV1/AISIH13/DIN1.2344 tool steel) corresponds to the H13 tool steel produced in the United States and the DIN1.2344 tool steel produced in Germany. Therefore, most of the thermophysical parameters employed in this paper are adopted with reference to different types of H13 steel of the same material [8]. The specific parameters are shown in Table 3, and these parameters are related to temperature. For the sake of accuracy, the transition from solidus to liquidus is considered to be a paste region in a certain temperature range. The melting temperature is assumed to be the average of the solidus and liquidus temperatures. In terms of fluid flow, the effective viscosity method allows the liquid phase to flow, but prevents any deformation of the solid portion. The actual dynamic viscosity at liquidus temperature is 5 mPa∙s. Because there is no fluid flow, the viscosity of the solid phase is considered to be infinite. In this study, the viscosity of the solid phase is set to at least 105 Pa∙s, which is several times that of the liquid phase. In addition, the dynamic viscosity decreases steadily from 105 Pa∙s in solid state to 5 mPa∙s in liquid state. In addition, the constructed viscosity corresponds to a “smooth” step function, which can not only describe the difference between liquid phase and solid phase, but also achieve numerical convergence. The rest of the properties of SKD61 materials are constant properties independent of temperature, and their specific values are shown in Table 3.

### 3.5. Boundary Conditions

When setting the boundary conditions of fluid heat transfer, laser heat source, thermal convection and thermal radiation act on boundary 2, as shown in Equation (12):(12)$$-k\nabla T=f\left(x_{1}\right)I+h\left(T-T_{a}\right)+\varepsilon \sigma \left(T^{4}-T_{a}^{4}\right)$$
(13)$$f\left(x_{1}\right)=\left\{\begin{array}{l} 0\;\;\;\;\;\;\left|x_{1}\right|\geq x_{0}\\ 1\;\;\;\;\;\;\;\left|x_{1}\right|\leq x_{0} \end{array}\right.$$
(14)$$x_{1}=x-v_{speed}\cdot t-0.08$$
(15)$$I=\alpha \frac{P}{\pi r_{0}{}^{2}}$$

The first term on the right of Equation (12) represents the moving heat source, $x_{1}$ is the length from the center of the spot, and its relationship with time t is shown in Equation (14), and $v_{speed}$ is the laser scanning speed. $f\left(x_{1}\right)$ is the range of laser action, and the laser energy beyond the range of the spot diameter is regarded as being equal to 0. I is a laser heat source acting on the free surface in the form of heat flux, and the laser energy is in a top-hat distribution, as shown in Equation (15), where α is the absorptivity of the material, P is the laser power, and $r_{0}$ is the spot radius of the laser beam acting on the sample surface. The second term in the formula is the thermal convection caused by the temperature difference, h is the convective heat transfer coefficient, $T_{a}$ is the ambient temperature. The last item is thermal radiation, and represents the radiation from the surface to the environment, and σ stands for the Stefan–Boltzmann constant.

Boundaries 1 and 3 are subject to thermal convection and thermal radiation from the surface to the environment, as described by the following formula
(16)$$-k\nabla T=h\left(T-T_{a}\right)+\varepsilon \sigma \left(T^{4}-T_{a}^{4}\right)$$

Boundary 4 is considered to be thermally insulated, expressed as
(17)∇T=0

When setting the boundary conditions of laminar flow, boundary 2 is defined as a freely deformable surface. The force σ acting on the top of the molten pool can be expressed as
(18)$$\sigma =\sigma_{n}+\sigma_{t}$$
where $\sigma_{n}$ denotes the capillary force acting in the normal direction, and the capillary force is related to the curvature of the free surface profile, which can be expressed as curvature. $\sigma_{t}$ expresses the thermal capillary force acting in the tangent direction, which is the tangential force caused by the temperature gradient on the surface of the molten pool, as shown in Figure 10, and its governing equation is as follows:(19)$$\sigma_{n}=\kappa \gamma \cdot n$$
(20)$$\sigma_{t}=\frac{\partial \gamma }{\partial T}\nabla_{S}T\cdot t$$
where κ expresses the free surface curvature, γ is the surface tension coefficient, which can be written as a function related to temperature γ = $\gamma_{m}$ − (∂γ/∂T)($T-T_{m}$), $\gamma_{m}$ is the surface tension coefficient of pure metal, ∂γ/∂T is the surface tension temperature coefficient, and $\nabla_{S}T$ is the temperature gradient along the tangent direction of the surface, n and t are expressed as unit normal vector and unit tangential vector, and $\frac{\partial \gamma }{\partial T}$ denotes the surface tension temperature gradient.

The tangential force can be applied to boundary 2 by the Marangoni effect of the inherent multi-physical-field coupling interface of COMSOL, while the normal force can be realized as a weak contribution to the boundary. The second term on the right of the weak contribution is called the contour integral, and is equal to zero. The specific boundary conditions are shown in Table 4.
(21)$$\begin{array}{l} \int {}_{S}\tilde{u}\sigma_{n}dS=\int {}_{S}\kappa \gamma \tilde{u}\cdot ndS=\gamma \int {}_{S}\left(-\nabla \cdot n\right)\tilde{u}\cdot ndS\\ \;\;\;\;\;\;\;\;\;\;\;\;\;\;\;\;\;\;=-\gamma \left[\int {}_{\partial S}\left(\tilde{u}\cdot n\right)dI-\int {}_{S}\left(\nabla \tilde{u}\right)\cdot dS\right]\\ \;\;\;\;\;\;\;\;\;\;\;\;\;\;\;\;\;\;=\int {}_{S}\gamma \left(\nabla \tilde{u}\right)\cdot dS-\int {}_{\partial S}\gamma \left(\tilde{u}\cdot n\right)dI \end{array}$$

To avoid the mesh deformity caused by poor accuracy, the arbitrary Lagrangian–Euler (ALE) method is used to track the displacement of the molten pool interface. When using this method, the grid velocity $V_{mesh}$ on the free interface is equal to the material velocity calculated according to the momentum conservation equation, as shown in the following formula:(22)$$V_{mesh}\cdot n=u\cdot n$$

Boundaries 1, 3, and 4 are all non-slip walls, and their boundary conditions are as follows:(23)u·n=0

### 3.6. Meshing

The simulation model uses free triangle element for meshing. To track the deformation of the free surface more accurately, boundary 2 is calibrated by hydrodynamics and meshed with extremely fine size. Because the residual solution domain is not melted, the calculation time and cost can be saved by using ordinary physical calibration. The average quality of the grid is over 93%. Table 5 shows the parameters of the grid, and the grid division of the computing domain is shown in Figure 11. The complete grid of the whole solution domain consists of 14,357 domain elements and 665 boundary elements, and the total number of degrees of freedom is 75,050, including 29,384 internal degrees of freedom.

### 3.7. Solver Configuration

The simulation uses the direct linear system solver PARDISO in the full coupling method, which has the advantages of high computational efficiency, small memory consumption and supporting shared parallel computing. To improve the accuracy and convergence of the model, a relative tolerance of 0.005 is adopted.

## 4. Simulation Results and Discussion

### 4.1. Evolution of Surface Profile of Molten Pool

The action time of laser energy density in the simulation is 15 ms, and the cooling time is 5 ms. Figure 12 shows the temperature distribution in the molten pool without magnetic field (MF = 0 T) at 5 ms, 10 ms, 15 ms, and 20 ms. In addition, Figure 13 shows the temperature distribution in the molten pool with MF = 0.3 T at 5 ms, 10 ms, 15 ms, and 20 ms. The white lines and color legends represent the isotherms and temperature fields, respectively. It can be seen from Figure 12 and Figure 13 that the surface topography gradually becomes smooth after laser polishing. In addition, it is also obvious that there is no significant change in the temperature distribution of the model surface without magnetic field at t = 5 ms and t = 10 ms compared with MF = 0.3 T. However, there is a slight decrease in the maximum temperature of the model surface at a magnetic field strength of 0.3 T at t = 15 ms compared with MF = 0 T. The results show that the presence of the magnetic field has almost no effect on the temperature distribution of the molten pool. Compared with the magnetic field, heat conduction and heat convection play a more important role in the temperature distribution of the melt pool. Therefore, under the action of the magnetic field, the values for parameters such as the depth, width, and solidification time of the molten pool are basically the same as those for laser polishing without the magnetic field, so the microstructure and other related characteristics of the remelting layer will not change. This means that the original parameters of laser remelting can remain unchanged regardless of whether a steady magnetic field is applied or not, avoiding the time-consuming adjustment of process parameters.

Unlike the temperature distribution, the fluid velocity is sensitive to the magnetic field. Figure 14 shows the velocity on the longitudinal section of the molten pool. The black line, white line and color legend represent the melting temperature line, streamline, and velocity field, respectively. Convection caused by the Marangoni effect can be observed in the molten pool. However, under the application of a magnetic field, the Marangoni eddy current slows down in the molten pool, and the maximum moving velocity of the molten metal decreases slightly, with the maximum velocity decreasing from 0.44 m/s to 0.41 m/s, as shown in Figure 14. The main reason for this phenomenon is that the Lorentz force of the melt pool is related to the melt pool flow velocity. The higher the melt pool flow velocity, the more obvious the inhibition of the melt pool flow by the Lorentz force, so the maximum velocity of the melt pool flow decreases slightly. However, the Lorentz force is the resistance to the flow of molten pool, and leads to the maximum velocity of the melt pool flow being slightly reduced. Figure 15 compares the fluid velocity distribution on the surface of the molten pool with and without application of the magnetic field. The double peak is due to the formation of circulation at both ends of the melt pool under the action of laser energy (see Figure 14), the circulation on the left side reaches the maximum flow velocity at X = 400 mm, and a peak occurs, while the circulation flow is in the opposite direction on the right side, and the melt pool flow velocity reaches a peak in the opposite direction at X = 600 mm. The results presented in Figure 15 show that the fluid velocity in the molten pool is suppressed slightly when the magnetic field density is 0.3 T. It can be predicted that the fluid velocity in the molten pool will further decrease with increasing magnetic flux density.

The simulation results show that the steady magnetic field has a slight influence on the velocity distribution during the laser remelting process, but it can still maintain the heat transfer conditions during the melting process. This phenomenon shows that the change in the temperature field is almost negligible, and the steady magnetic field reduces the surface roughness by inhibiting the secondary flow of the molten pool.

Figure 16 shows the distribution of drive force of molten pool at t = 10 ms. It can be found from Figure 16 that the thermocapillary force exceeds the capillary force and the Lorentz force to dominate the molten pool, where 375 μm < r < 425 μm and 580 μm < r < 620 μm. In addition, the thermocapillary force is almost equal to the Lorentz force, where 330 μm < r < 375 μm. Moreover, it is obvious that the Lorentz force reaches its maximum and has the most obvious effect on the molten pool at X = 400 mm.

Comparing the surface morphology of the samples obtained via simulation with and without magnetic field. It can be seen that surface fluctuation is inevitable due to the periodic motion of the melt during the process of laser polishing. According to the governing equation (Equation (2)), the normal component of the deformation velocity of the mesh at the interface is equal to the normal fluid velocity, so the ALE method can be used to accurately describe the surface topography. Due to the flow being inhibited by the magnetic field, the fluctuation of the molten pool becomes smaller, leading to the velocity of the molten pool decreasing. Finally, the surface morphology of the model changes. The inhibition of the magnetic field is mainly due to the fact that the directions of Lorentz force and fluid velocity are always opposite to one another. The simulation results show that the steady magnetic field has little effect on heat conduction, meaning that despite the application of the magnetic field, the solidification time of the molten pool remains almost constant, almost always solidifying at t = 20 ms (see Figure 12). The results show that the surface fluctuation is suppressed slightly during the process of magnetic field-assisted continuous laser polishing.

Figure 17 shows the simulated surface profile of the unpolished surface, polished without magnetic field and polished with MF = 3 T, at t = 20 ms. It can be observed from Figure 17 that laser polishing effectively improves the irregular peaks and valleys of the matrix, effectively making them smooth. However, the magnetic field has little effect on the surface morphology. In addition, there is a slight decrease in the height of the peaks after laser polishing with MF = 3 T compared with polishing without the magnetic field. As can be seen in Figure 17, the peaks are not equal, mainly due to the melt pool flow velocity being the highest and the Lorentz force having an inhibiting effect on the fluid flow at X = 400 mm, resulting in unequal surface peaks.

### 4.2. Experimental Verification

To verify the simulation results, laser polishing experiments are carried out using a continuous fiber laser with a maximum power of 1000 W. In addition, the morphology of the cross-sectional molten pool is observed using a scanning electron microscope (model: ZEISS Gemini 300, Gina, Germany). A comparison of the simulated and experimental melt pool morphology is presented in Figure 18. The results show the depths of the molten pool obtained experimentally and via simulation are 56.52 μm and 65 μm, respectively. Compared with the experimental results, the simulation error is 15.0%, illustrating the accuracy of the simulation model.

## 5. Conclusions

The effects of energy density and magnetic field intensity on surface roughness were studied. Whether the magnetic field was added or not, the surface roughness was improved. Under the optimal parameters of magnetic field-assisted laser polishing, the surface roughness of the material was reduced to 0.607 μm, representing a reduction rate of up to 90.5%.

The underlying mechanism of the effect of the magnetic field on the evolution of the surface morphology and the flow behavior of the molten pool was simulated. The numerical simulation results showed that the Lorentz force generated by the molten pool flow inhibited the secondary overflow of the molten pool, thereby reducing the surface roughness of the material slightly. In addition, the correctness of the numerical model was verified by the experiments. The depth of the actual weld pool was 56.52 μm, and the depth of the simulated weld pool was 65 μm. Compared with the molten pool depth obtained during the experiment, the simulation error was 15.0%.

## Figures and Tables

**Figure 1 micromachines-13-01493-f001:**
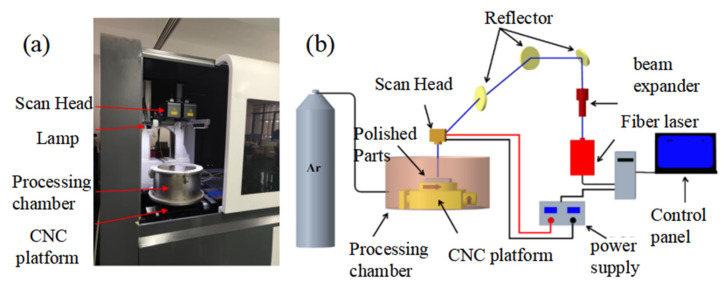
Laser polishing test equipment and schematic diagram: (**a**) test device (**b**) principle of test device.

**Figure 2 micromachines-13-01493-f002:**
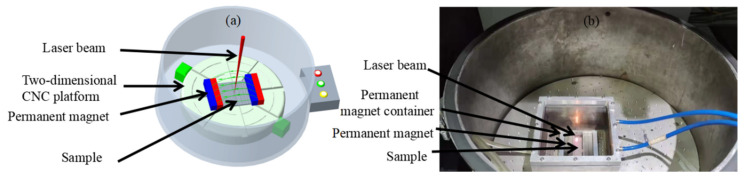
Test principle and installation diagram of magnetic field-assisted laser polishing: (**a**) principle of magnetic field test device; (**b**) magnetic field auxiliary test device.

**Figure 3 micromachines-13-01493-f003:**
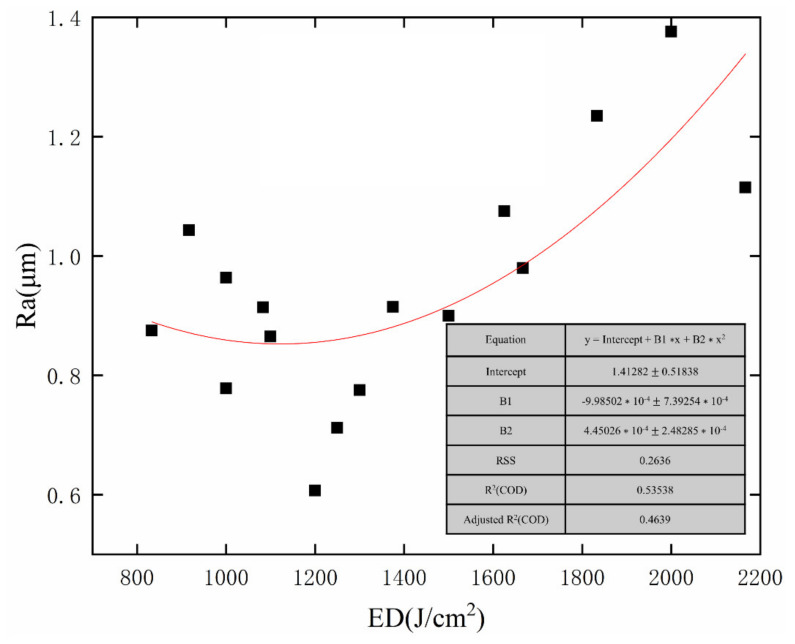
The relationship between energy density (ED) and surface roughness (Ra) assisted by magnetic field.

**Figure 4 micromachines-13-01493-f004:**
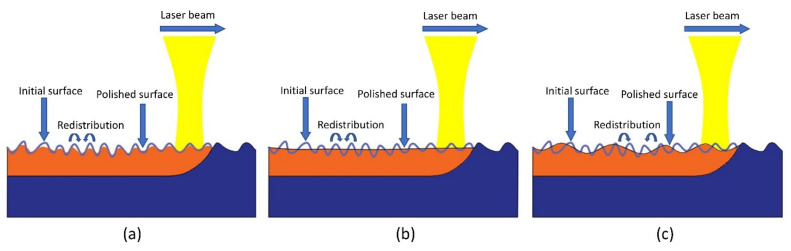
Mechanism diagram of laser polishing: (**a**) incomplete melting; (**b**) shallow surface melting; (**c**) surface over-melting.

**Figure 5 micromachines-13-01493-f005:**
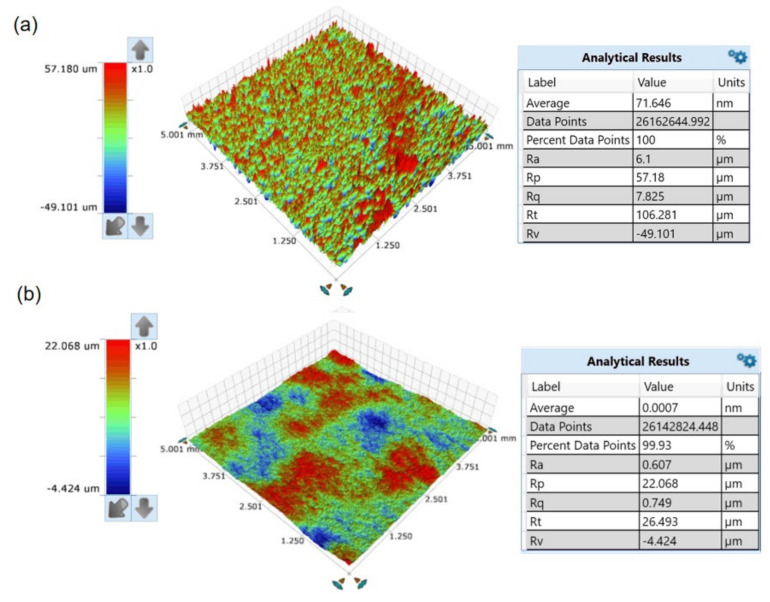
The three-dimensional morphology of the measured sample: (**a**) initial surface; (**b**) sample 11 surface with P = 180 W, V = 50 mm/s and MF = 0.3 T.

**Figure 6 micromachines-13-01493-f006:**
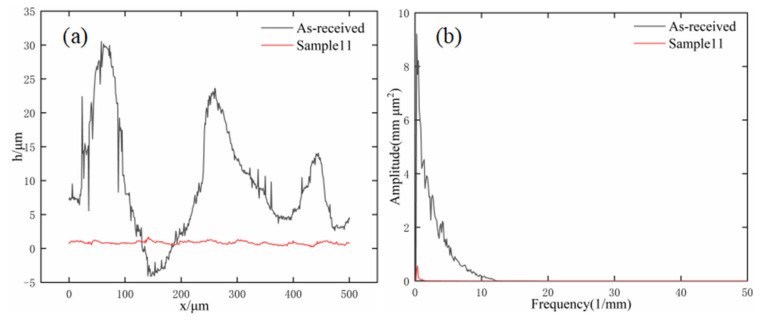
Analysis of surface topography and power spectral density before and after polishing: (**a**) surface topography; (**b**) power spectral density analysis.

**Figure 7 micromachines-13-01493-f007:**
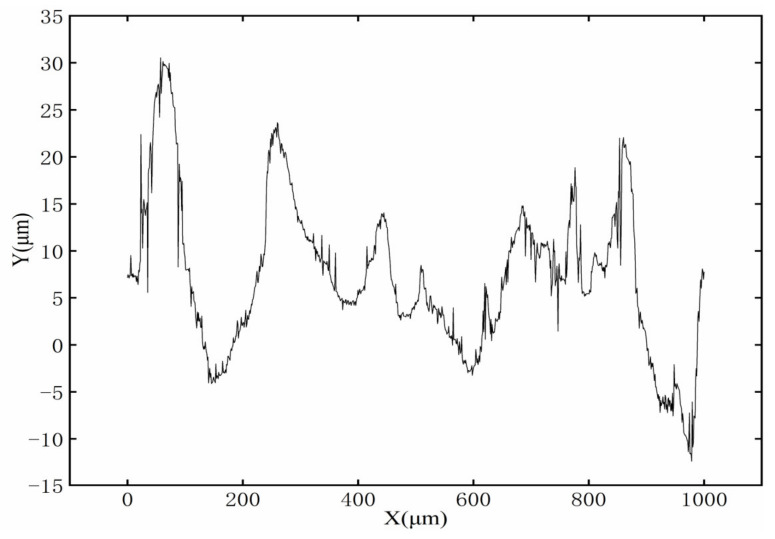
Initial surface profile.

**Figure 8 micromachines-13-01493-f008:**
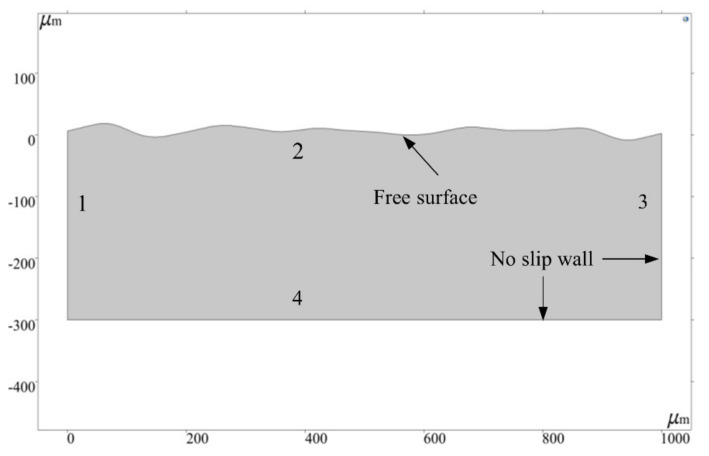
Model geometry.

**Figure 9 micromachines-13-01493-f009:**
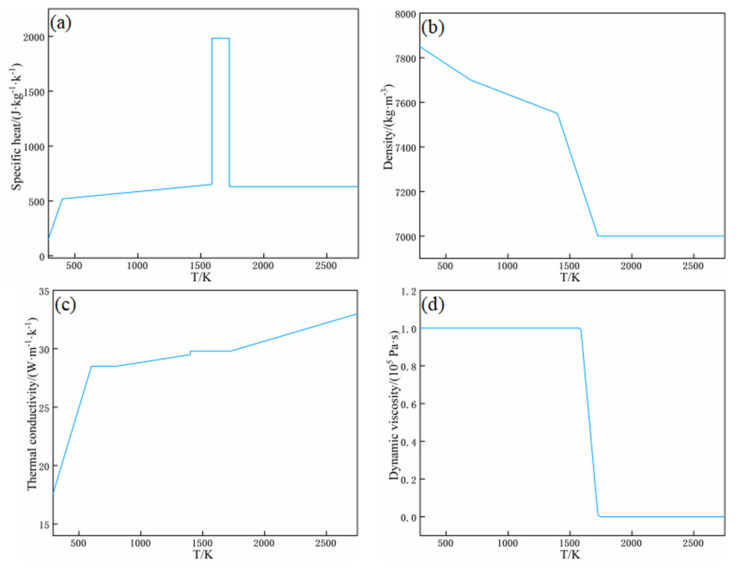
Thermophysical parameters of the materials: (**a**) specific heat capacity; (**b**) density; (**c**) thermal conductivity; (**d**) dynamic viscosity.

**Figure 10 micromachines-13-01493-f010:**
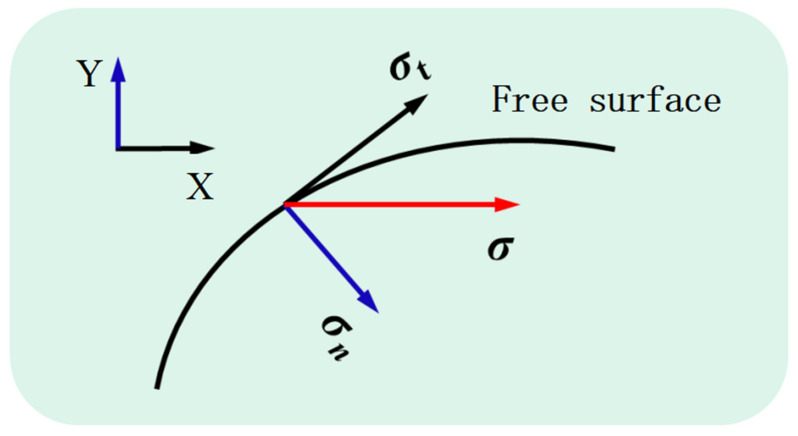
Action of tangential force and normal force along the free surface.

**Figure 11 micromachines-13-01493-f011:**
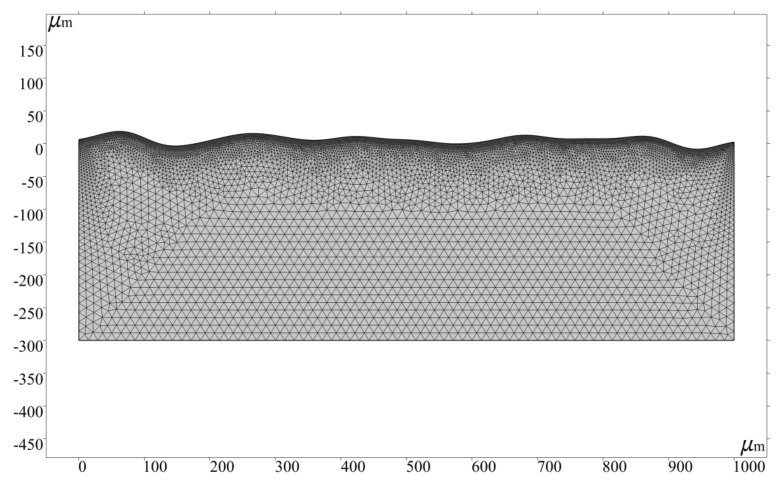
Grid.

**Figure 12 micromachines-13-01493-f012:**
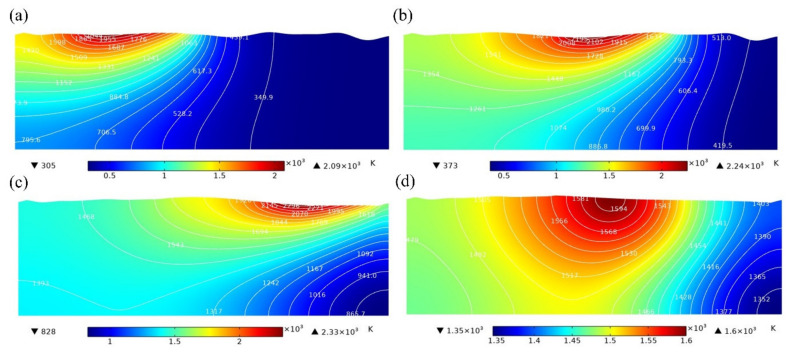
The temperature distribution of molten pool without magnetic field at (**a**) t = 5 ms, (**b**) t = 10 ms, (**c**) t = 15 ms, and (**d**) t = 20 ms.

**Figure 13 micromachines-13-01493-f013:**
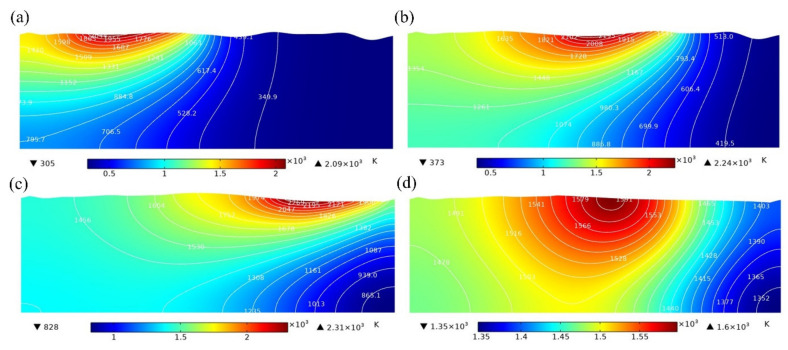
The temperature distribution of molten pool with MF = 0.3 T at (**a**) t = 5 ms, (**b**) t = 10 ms, (**c**) t = 15 ms, and (**d**) t = 20 ms.

**Figure 14 micromachines-13-01493-f014:**
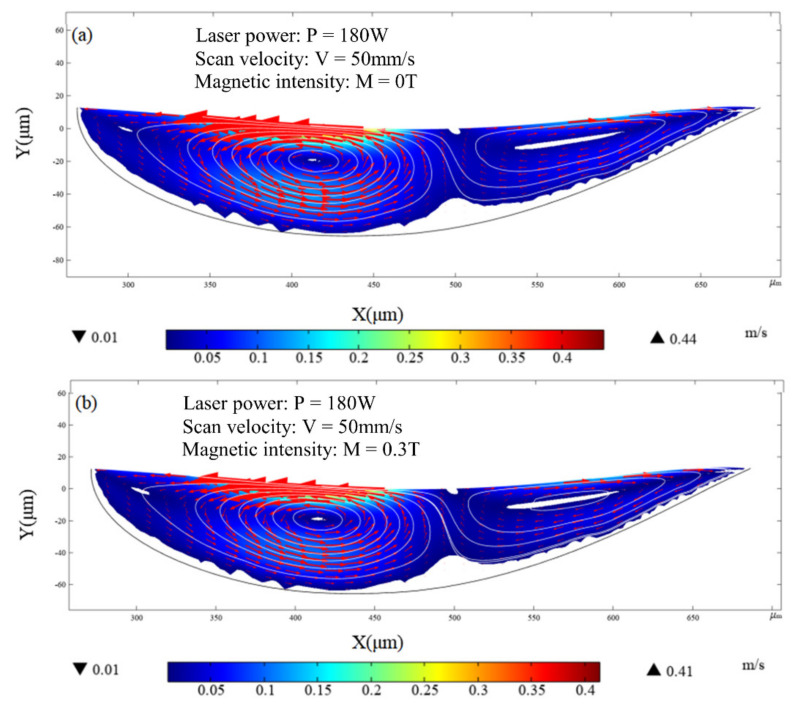
Velocity distribution of fluid in molten pool at t = 10 ms: (**a**) without magnetic field (MF = 0 T); (**b**) with magnetic field (MF = 0.3 T).

**Figure 15 micromachines-13-01493-f015:**
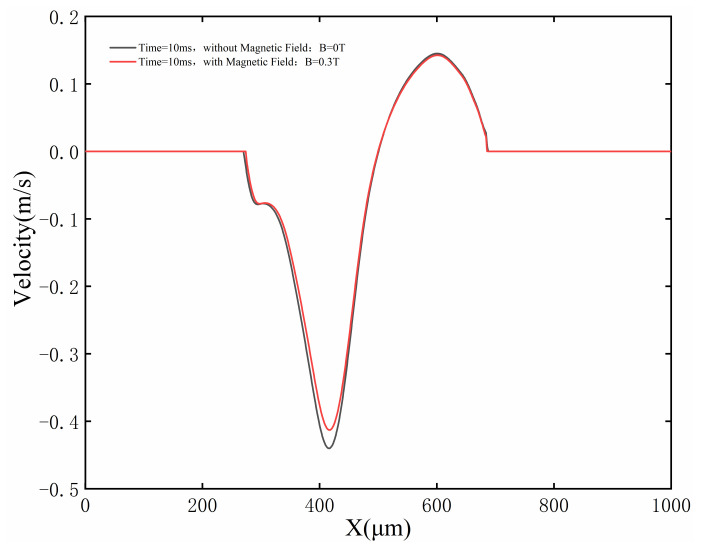
The velocity at the top of the molten pool with and without application of magnetic field at t = 10 ms.

**Figure 16 micromachines-13-01493-f016:**
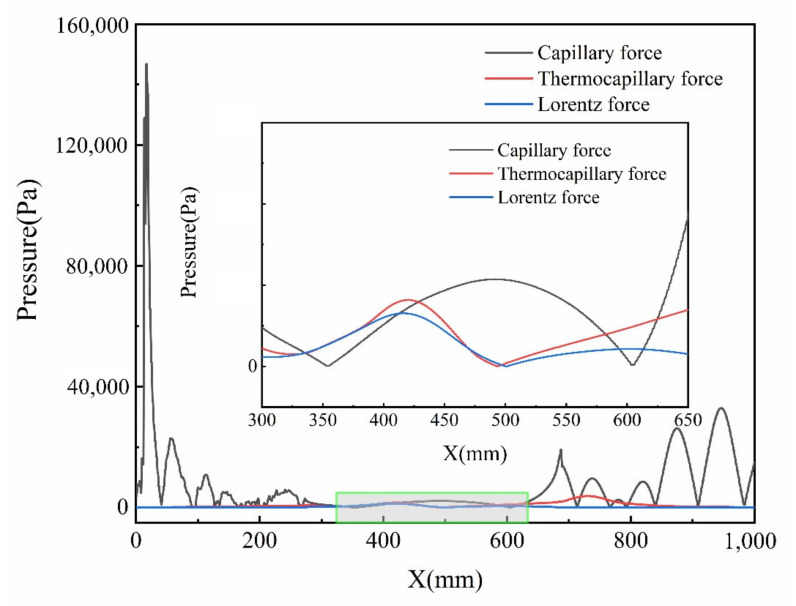
Variation in capillary force, thermocapillary force and Lorentz force at t = 10 ms.

**Figure 17 micromachines-13-01493-f017:**
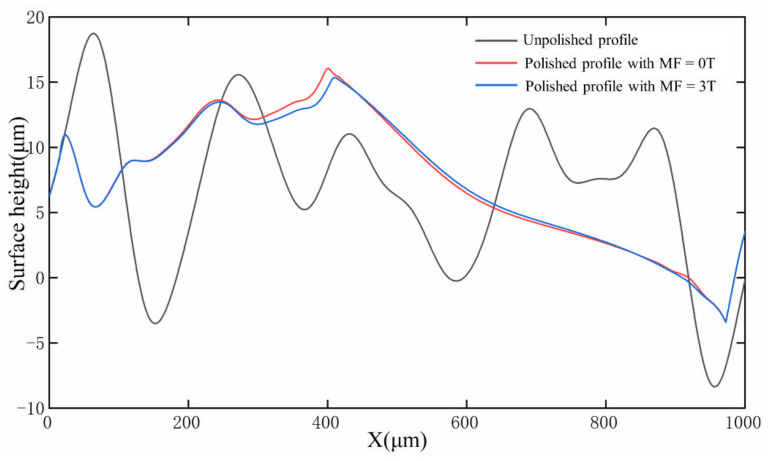
Morphology of the top of the molten pool at t = 20 ms.

**Figure 18 micromachines-13-01493-f018:**
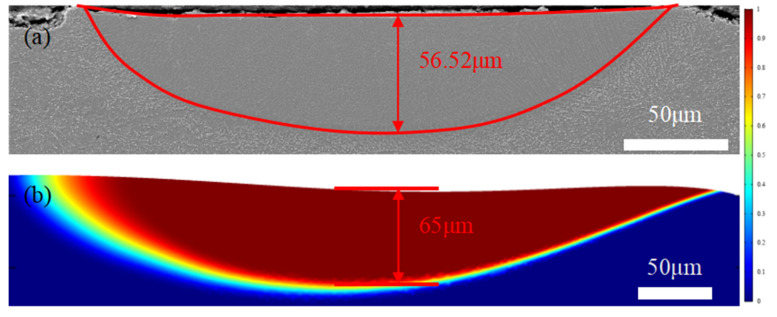
Experimental verification: (**a**) actual weld pool shape and (**b**) simulated shape of molten pool at t = 14 ms.

**Table 1 micromachines-13-01493-t001:** Chemical composition of SKD61 die steel.

Chemical Element	C	Si	Mn	P	S	Cr	Mo	Ni	Cu	V	Fe
Mass fraction (%)	0.3–0.45	0.8–1.2	0.2–0.5	0.03	0.03	4.75–5.5	1.1–1.75	0.3	0.25	0.8–1.2	≥88

**Table 2 micromachines-13-01493-t002:** L16(4^4^) magnetic field-assisted CW laser orthogonal test table.

Specimen Code	P (W)	V (mm/s)	D (mm)	MF (T)	ED (J/cm^2^)	Ra (μm)
As-received	–	–	–	–	–	6.1
1	150	30	0.03	0	1666.667	0.980
2	150	40	0.05	0.3	1250.000	0.712
3	150	50	0.07	0.6	1000.000	0.778
4	150	60	0.09	0.9	833.333	0.875
5	165	30	0.05	0.6	1833.333	1.235
6	165	40	0.03	0.9	1375.000	0.915
7	165	50	0.09	0	1100.000	0.865
8	165	60	0.07	0.3	916.667	1.043
9	180	30	0.07	0.9	2000.000	1.376
10	180	40	0.09	0.6	1500.000	0.901
11	180	50	0.03	0.3	1200.000	0.607
12	180	60	0.05	0	1000.000	0.964
13	195	30	0.09	0.3	2166.667	1.115
14	195	40	0.07	0	1625.000	1.075
15	195	50	0.05	0.9	1300.000	0.775
16	195	60	0.03	0.6	1083.333	0.914

**Table 3 micromachines-13-01493-t003:** Simulation parameters of SKD61.

Property (Units)	Symbol	Value
Liquidus temperature (K)	$T_{l}$	1727
Solidus temperature (K)	$T_{s}$	1588
Melting temperature (K)	$T_{m}$	1657.5
Reference temperature (K)	$T_{ref}$	1727
Latent heat of fusion (J/kg)	$L_{f}$	2.8 × 10^5^
Surface tension of pure metal (N/m)	$\gamma_{m}$	1.909
Thermal expansion coefficient (1/K)	β	1 × 10^−4^
Convective coefficient (W/(m^2^·K))	h	10
Emissivity	ε	0.5
Constant of surface tension gradient (N/(m·K))	$A_{\gamma }$	5.2 × 10^−4^

**Table 4 micromachines-13-01493-t004:** Boundary conditions.

Physics	Physical Condition	Boundary	Boundary Condition
Heat transfer	Laser irradiation	2	Heat flux
	Natural convection	1,2,3	Convection
	Radiation	1,2,3	Diffuse surface
	Insulation	4	Thermal insulation
Fluid flow	Normal stress	2	Weak contribution
	Tangential stress	2	Marangoni effect
	Wall	1,3,4	No-slip wall
Magnetic field	Insulation	1,2,3,4	Electrical insulation
	Insulation	1,2,3,4	Magnetic insulation
Moving mesh	Fixed boundary	1,3,4	Prescribed displacement
	Free deformation	2	Prescribed displacement

**Table 5 micromachines-13-01493-t005:** Grid size parameters.

Parameter (Unit)	Top Layer	The Rest
Maximum element size (μm)	2	14.3
Minimum element size (μm)	0.1	0.638
Maximum element growth rate	1.05	1.15
Curvature factor	0.2	0.3

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
