# Peer review of "Laser Polishing Die Steel Assisted by Steady Magnetic Field"

_micromachines, 2022, doi:10.3390/mi13091493_

Round 1
Reviewer 1 Report
1. The Chinese characters shoud be changed to English characters in Fig. 3 and Fig. 14. There are only “Fig.” without figure number in the manuscript, such as in line 222, line 548. Where are the images of Fig. 3.3 (line 351) and Fig. 3.9 (line 410)? There are many errors in Engligh grammer. Please take a careful look throughout the manuscript to correct them.
2. Introduction
The problem statement is poorly described. The current state of the art in laser machining assisted by magnetic field is not even discussed in the literature review. The authors should focus on these aspects in the introduction to bring clarity to the need for such researches. Moreover, there are no references about the presentation of the SKD61 steel.
3. Materials and Methods:
1) Did the water surface exceed the upper workpiece surface during laser polishing? If so, how should the influence of laser energy loss on processing be considered?
2) How do gravity, buoyancy, surface tension, thermocapillary force and Lorentz force affect the surface morphology? The authors should give a force diagram to explain the phenomenon.
3) The authors should provide more details about the types, manufacturers and production places of many relevant experimental equipment, such as white light ingerferometer, Tesla meter.
4. Model building:
1) The boundaries 1-4 are not marked in Fig. 8.
2) The experimental temperature distribution with and without magenitc field during laser polishing should be measured to compare with the numerical simulations.
3) How should the velocity distribuion indicate when the magnetic field density is greater than 0.3 T?
4) More experimental results with different machining parameters should be displayed to verify the numerical simulations.
5) Why the heat conduction plays a more significant role than heat convection in the molten pool?
5. What is the innovation of the manuscript as compared with the reference “[18] Wang L, Yao J,Hu Y, et al. Suppression effect of a steady magnetic field on molten pool during laser remelting[J].Applied 612 Surface Science, 2015,351:794–802.”
Author Response
Micromachines-1847487
Response to Reviewer
Dear Reviewer,
Thank you for giving us the opportunity to submit a revised draft of the manuscript “Laser polishing die steel assisted by steady magnetic field” for publication in the Journal of Micromachines. We appreciate the time and effort that you dedicated to providing feedback on our manuscript and are grateful for the insightful comments on and valuable improvements to our paper.
We have incorporated most of the suggestions made by the reviewers. Those changes are highlighted within the manuscript. Please see below, in blue, for a point-by-point response to the reviewers’comments and concerns. All page numbers refer to the refer to the revised manuscript file with tracked changes.
- The Chinese characters shoud be changed to English characters in Fig. 3 and Fig. 14. There are only “Fig.” without figure number in the manuscript, such as in line 222, line 548. Where are the images of Fig. 3.3 (line 351) and Fig. 3.9 (line 410)? There are many errors in Engligh grammer. Please take a careful look throughout the manuscript to correct them.
Responds: Thank you for this suggestion. I have revised Figure 3 and Figure 4 in the paper and added the figure numbers in line 207, line 595. Moreover, Fig. 3.3 (line 351) and Fig. 3.9 (line 410) in the paper are Table3 (line 328) and Fig. 10(line387), respectively, which I have revised over in the paper.
2 Introduction
The problem statement is poorly described. The current state of the art in laser machining assisted by magnetic field is not even discussed in the literature review. The authors should focus on these aspects in the introduction to bring clarity to the need for such researches. Moreover, there are no references about the presentation of the SKD61 steel.
Responds: Thank you for this suggestion.I have revised the introduction section in my paper and added the relevant references.
3 Material and Methods
1) Did the water surface exceed the upper workpiece surface during laser polishing? If so, how should the
influence of laser energy loss on processing be considered?
Responds: The height of the water is over the upper surface, but the water is only added to the permanent magnet container and does not come into contact with the workpiece. The water is used to cool down the permanent magnets and does not cause a loss of laser energy.
2) How do gravity, buoyancy, surface tension, thermocapillary force and Lorentz force affect the surface
morphology? The authors should give a force diagram to explain the phenomenon.
Responds: Thank you for this suggestion. A force diagram including capillary force, thermocapillary force and Lorentz force is given in Fig. 16.(line503)
- The authors should provide more details about the types, manufacturers and production places of many relevant experimental equipment, such as white light ingerferometer, Tesla meter.
Responds: Thank you for this suggestion. I have added the model number of the relevant equipment in my paper.(line 131) (line187)
4 Mold building
1) The boundaries 1-4 are not marked in Fig. 8.
Responds: Thank you for this suggestion. I have marked the boundaries 1-4 in Figure 8. (line 315)
2) The experimental temperature distribution with and without magenitc field during laser polishing should be measured to compare with the numerical simulations.
Responds: Thank you for this suggestion. I am sorry, due to the experimental conditions, The group does not have a thermal imager and is unable to check the temperature of the workpiece in real time.
3) How should the velocity distribuion indicate when the magnetic field density is greater than 0.3 T?
Responds: Thank you for this suggestion. Under the same parameters, there is no significant difference in the velocity distribution of the melt pool when the magnetic field strength is 0.6T or 0.9T.
4) More experimental results with different machining parameters should be displayed to verify the numerical simulations.
Responds: Thank you for this suggestion. By magnetic field assisted laser polishing L16(44) orthogonal experiments, 16 sets of experimental results were obtained, and the best one was selected for verification.
5) Why the heat conduction plays a more significant role than heat convection in the molten pool?
Responds: Thank you for this suggestion. I am really sorry for the semantic change due to a translation error. I have revised the paper and the revised statement is: Compared with magnetic field, heat conduction and heat convection play a more important role in the melt pool temperature distribution.(line 443)

Reviewer 2 Report
A well-written and organised manuscript which is worth publishing after addressing the following issues:
-FEA simulation of the physical problem presented in this paper, as stated in its introduction, has already been done, and the claim that more publication is needed is not enough to show the novelty of your paper. What is special about your work that no one has done before?
-The boundaries are numbered 1-4 but not shown on the computational domain, please do show those in one of the relevant figures.
-Boundary 4 is selected to be thermally insulated, which is not realistic, all the boundaries should be set to the ambient temperature, because T at the boundary is around 700 -1200 K which is significant and if an adiabatic condition is used, then the heat will be trapped in this 1 mm3 domain. The more realistic set up to either take a much larger area and allow the adjacent material to absorb part of the heat in the weld pool, or set the boundaries to ambient temperature to allow a more realistic heat dissipation in the model.
-is mentioned that Fig 15 peaks indicate heat accumulation due to the Marangoni effect. This is very vague and any reader would not be able to make sense of this claim unless it is explained mathematically or in any other way.
- why the peaks are not equal and biased to one side? Explain the values.
-Does the value of the velocity proves that the flow is laminar? please evidence this.
-why only 0.3 T is used? Why not try different levels?
-what is the relative tolerance mentioned in L452?
-Different frames showing the evolution of the simulation need to be provided.
-The solidification rate and the temperature distribution need to be shown at different time frames including the start and the end of the simulation.
-it is claimed in L479 – L 500 that the fluid velocity is “significantly decreased”, however it is only reduced by approx 7%, from 0.44 to 0.421. this claim is not entirely true.
-Many claims in the paper need to be quantified, such as the cooling rate in L525 etc.
-Show that the fluctuation of the surface is reduced with/without the magnetic field.
-In Fig 16, there is not a significant difference in the morphology between with/without magnetic field.
-Fig 16, Why the roughness was a bit different only in the centre?
-Fig 16 shows the time of 20 ms, while Fig 17 shows the comparison at the time 40 ms, what happens at the different time frames? Or at least when the whole process finished and the sample cooled. Is 40 ms the last timeframe?
-I think the results/conclusion from this research need to be different, because the magnetic field did not make a significant difference, especially a little bit reducing the height of the profile, but not necessarily the roughness. The roughness was mainly reduced because of the laser melting.
The manuscript needs to be significantly reviewed before being accepted
Author Response
Micromachines-1847487
Response to Reviewer
Dear Reviewer,
Thank you for giving us the opportunity to submit a revised draft of the manuscript “Laser polishing die steel assisted by steady magnetic field” for publication in the Journal of Micromachines. We appreciate the time and effort that you dedicated to providing feedback on our manuscript and are grateful for the insightful comments on and valuable improvements to our paper.
We have incorporated most of the suggestions made by the reviewers. Those changes are highlighted within the manuscript. Please see below, in blue, for a point-by-point response to the reviewers’comments and concerns. All page numbers refer to the refer to the revised manuscript file with tracked changes.
1 FEA simulation of the physical problem presented in this paper, as stated in its introduction, has already been done, and the claim that more publication is needed is not enough to show the novelty of your paper. What is special about your work that no one has done before?
Responds: Thank you for this suggestion. None of these scholars has conducted an in-depth study of the evolution of the melt pool hydrodynamics, electromagnetism and surface topography in the molt pool region. I have studied these aspects.
2 The boundaries are numbered 1-4 but not shown on the computational domain, please do show those in one of the relevant figures.
Responds: Thank you for this suggestion. I've revised the diagram in the paper.(line 315)
3 Boundary 4 is selected to be thermally insulated, which is not realistic, all the boundaries should be set to the ambient temperature, because T at the boundary is around 700 -1200 K which is significant and if an adiabatic condition is used, then the heat will be trapped in this 1 mm3 domain. The more realistic set up to either take a much larger area and allow the adjacent material to absorb part of the heat in the weld pool, or set the boundaries to ambient temperature to allow a more realistic heat dissipation in the model.
Responds: I refer to Zhang Chi's paper, whose boundary4 is also set to the insulation boundary.
Zhang C, Zhou J, Shen H. Role of capillary and thermocapillary forces in laser polishing of Metals [J]. Journal of Manufacturing Science and Engineering, 2017, 139.
4 is mentioned that Fig 15 peaks indicate heat accumulation due to the Marangoni effect. This is very vague and any reader would not be able to make sense of this claim unless it is explained mathematically or in any other way.
Responds: The double peak is due to the formation of circulation at both ends of the melt pool under the action of laser energy (see fig. 14), the circulation on the left side reaches the maximum flow velocity at X=400mm and a peak occurs, while the circulation is in the opposite direction on the right side, and the melt pool flow velocity reach a peak in the opposite direction at X=600mm. And I've revised it in the paper.(line 475)
5 why the peaks are not equal and biased to one side? Explain the values.
Responds: As is shown in Fig. 17, the peaks are not equal mainly due to the melt pool flow velocity is the largest and the Lorentz force has inhibiting effect on the fluid flow at X=400mm, resulting in unequal surface peaks.
Biased to one side is due to the high velocity of the circulation flow on the left side of the melt pool leads to a large amount of material flowing to the left side, resulting in the peak biased to the left. (see Fig. 14). (line 525)
6 Does the value of the velocity proves that the flow is laminar? please evidence this.
Responds: According to the Reynolds number calculation formula: Re=, whereis the density of fluid, is the velocity of fluid, is the viscosity of fluid, is the feature length. The model Reynolds number is calculated to be 31.9 in this simulation model, which is much less than 2000, so it is laminar flow.
7 why only 0.3 T is used? Why not try different levels?
Responds: Thank you for this suggestion. The orthogonal experiment resulted in the lowest surface roughness at 0.3T, so only 03T was selected.。
8 what is the relative tolerance mentioned in L452?
Responds: The relative tolerance is the allowable deviation range, the smaller the relative tolerance, the more accurate the simulation results, the relative tolerance is 0.005 in this paper. (line 429)
9 Different frames showing the evolution of the simulation need to be provided.
Responds: Thank you for this suggestion. I have added the temperature distribution at different moments, including the beginning and the end of the simulation in Figure 12. (line 453), (line 456)
10 The solidification rate and the temperature distribution need to be shown at different time frames including the start and the end of the simulation.
Responds: Thank you for this suggestion. I have added the temperature distribution at different moments, including the beginning and the end of the simulation in Figure 12. (line 453), (line 456)
11 it is claimed in L479 – L 500 that the fluid velocity is “significantly decreased”, however it is only reduced by approx 7%, from 0.44 to 0.421. this claim is not entirely true.
Responds: Thank you for this suggestion.The magnetic field has a slight effect on fluid flow velocity. And I have revised it in the paper. (line 481)
12 Many claims in the paper need to be quantified, such as the cooling rate in L525 etc.
Responds: Thank you for this suggestion. I'm really sorry for the semantic change due to a translation error. I have revised the paper and the revised statement is: The simulation results show that the steady magnetic field has little effect on the heat conduction, which means that despite the magnetic field, the solidification time of the molten pool is almost constant. Almost always solidifies at t=20ms (see Fig. 12). (line 516)
13 Show that the fluctuation of the surface is reduced with/without the magnetic field.
Responds: It is obvious from Fig. 17 that the fluctuation of the sample surface is significantly reduced after laser polishing with/without the magnetic field. (line 529)
14 In Fig 16, there is not a significant difference in the morphology between with/without magnetic field.
Responds: The greater impact on the surface is the laser energy, the magnetic field only plays an auxiliary role, the magnetic field can reduce the secondary overflow generated by the laser polishing, and the magnetic field can further slightly reduce the surface roughness of the sample. (line 529)
15 Fig. 16, Why the roughness was a bit different only in the centre?
Responds: The main reason is that the Lorentz force is related to the flow speed of the melt pool, the greater the flow speed the greater the Lorentz force. Combined with velocity distribution in Fig. 16, it can be seen that the melt pool flow speed is the largest in X= 400mm, the Lorentz force has the greatest impact, so it will be a bit different in X=400mm. (line 529)
16 Fig 16 shows the time of 20 ms, while Fig 17 shows the comparison at the time 40 ms, what happens at the different time frames? Or at least when the whole process finished and the sample cooled. Is 40 ms the last timeframe?
Responds: The 20ms is the total simulation time in Fig. 16. Combining Fig. 12 and Fig. 13, it can be found that the melt pool is completely solidified at this time. Fig. 17 shows the surface topography at 14ms instead of 40ms, the melt pool is in a completely stable state at this time. Therefore, the simulated melt pool at 14 ms was chosen to compare with the experimental melt pool.
17 I think the results/conclusion from this research need to be different, because the magnetic field did not make a significant difference, especially a little bit reducing the height of the profile, but not necessarily the roughness. The roughness was mainly reduced because of the laser melting.
Responds: Thank you for this suggestion. The reduction in roughness is mainly due to laser melting indeed. As can be seen from Fig. 17, there is a slight decrease in the height of the peaks after laser polishing with a magnetic field compared to the surface morphology without magnetic field. And I have revised it in the paper. (line 553)

Round 2
Reviewer 1 Report
1. There are many grammatical errors, punctuation marks and reference marks in the manuscript. Please take a careful look throughout the manuscript to correct them.
2. Is the laser energy with Gaussian beam or flat-topped beam distribution during the experiments?
3. The resolution of Figure 4 is very ambiguous, which maybe does not meet the requirements of publication.
4. Why is the minimum roughness given in the manuscript with 0.607 μm and that given with 0.681 μm in Figure 5b? Please provide the experimental parameters to obtain the roughness value in Figure 5b.
5. Where is the reference 24 cited in the manuscript?
6. The abstract should be modified according to the numerical simulations and experimental results.
Author Response
Micromachines-1847487
Response to Reviewers
Dear Reviewer,
Thank you for giving us the opportunity to submit a revised draft of the manuscript “Laser polishing die steel assisted by steady magnetic field” for publication in the Journal of Micromachines. We appreciate the time and effort that you dedicated to providing feedback on our manuscript and are grateful for the insightful comments on and valuable improvements to our paper.
We have incorporated most of the suggestions made by the reviewers. Those changes are highlighted within the manuscript. Please see below, in blue, for a point-by-point response to the reviewers’ comments and concerns. All page numbers refer to the refer to the revised manuscript file with tracked changes.
Point 1: There are many grammatical errors, punctuation marks and reference marks in the manuscript. Please take a careful look throughout the manuscript to correct them.
Responds 1: Thank you for this suggestion. I have revised it in the paper.
Point 2: Is the laser energy with Gaussian beam or flat-topped beam distribution during the experiments?
Responds 2: The laser energy distribution used in the experiment is flat-topped beam distribution. (line 104)
Point 3: The resolution of Figure 4 is very ambiguous, which maybe does not meet the requirements of publication.
Responds 3: Thank you for this suggestion. I have revised it in the paper. (line 185)
Point 4: Why is the minimum roughness given in the manuscript with 0.607μm and that given with 0.681μm in Figure 5b? Please provide the experimental parameters to obtain the roughness value in Figure 5b.
Responds 4: I am really sorry for the surface topography pictures was wrong in Fig. 5b. I provide the correct surface topography image and the experimental parameters in Figure 5b. (line 215)
Point 5: Where is the reference 24 cited in the manuscript?
Responds 5: I am really sorry for missing this reference. And I have revised it in the paper. (line 613)
Point 6: The abstract should be modified according to the numerical simulations and experimental results.
Responds 6: Thank you for this suggestion. I have revised it in the paper.
